# The Solidification of Lead-Zinc Smelting Slag through Bentonite Supported Alkali-Activated Slag Cementitious Material

**DOI:** 10.3390/ijerph16071121

**Published:** 2019-03-28

**Authors:** Yanhong Mao, Faheem Muhammad, Lin Yu, Ming Xia, Xiao Huang, Binquan Jiao, YanChyuan Shiau, Dongwei Li

**Affiliations:** 1State Key Laboratory for coal mine disaster dynamics and control, Chongqing University, Chongqing 400044, China; 20162002020t@cqu.edu.cn (Y.M.); I1500208@cqu.edu.cn (F.M.); xiamingchn@126.com (M.X.); 2College of Resource and Environmental Science, Chongqing University, Chongqing, 400044, China; yl_518@139.com; 3Department of Landscape Architecture, Chung Hua University, No. 707, Wufu Rd., Sec. 2, Hsinchu 30012, Taiwan

**Keywords:** alkali activated slag, lead-zinc smelting slag, bentonite, solidification/stabilization

## Abstract

The proper disposal of Lead-Zinc Smelting Slag (LZSS) having toxic metals is a great challenge for a sustainable environment. In the present study, this challenge was overcome by its solidification/stabilization through alkali-activated cementitious material i.e., Blast Furnace Slag (BFS). The different parameters (water glass modulus, liquid-solid ratio and curing temperature) regarding strength development were optimized through single factor and orthogonal experiments. The LZSS was solidified in samples that had the highest compressive strength (after factor optimization) synthesized with (AASB) and without (AAS) bentonite as an adsorbent material. The results indicated that the highest compressive strength (AAS = 92.89MPa and AASB = 94.57MPa) was observed in samples which were prepared by using a water glass modulus of 1.4, liquid-solid ratio of 0.26 and a curing temperature of 25 °C. The leaching concentrations of Pb and Zn in both methods (sulfuric and nitric acid, and TCLP) had not exceeded the toxicity limits up to 70% addition of LZSS due to a higher compressive strength (>60 MPa) of AAS and AASB samples. While, leaching concentrations in AASB samples were lower than AAS. Conclusively, it was found that the solidification effect depends upon the composition of binder material, type of leaching extractant, nature and concentration of heavy metals in waste. The XRD, FTIR and SEM analyses confirmed that the solidification mechanism was carried out by both physical encapsulation and chemical fixation (dissolved into a crystal structure). Additionally, bentonite as an auxiliary additive significantly improved the solidification/stabilization of LZSS in AASB by enhancing the chemical adsorption capacity of heavy metals.

## 1. Introduction

Nowadays, the increase in hazardous wastes, which contains various heavy metals due to industrialization has attracted the attention of scientists regarding its proper disposal. Lead-Zinc Smelting Slag (LZSS) is an industrial waste produced during the smelting of lead (Pb) and zinc (Zn) from their ores. It is considered hazardous waste as to it contains a number of heavy metals including Pb, Zn, Cu, Mn, Cr, Cd and Ni etc. In addition, this kind of waste having heavy metals make it prone to leaching which results in the contamination of the ecosystem and ultimately threatens human health [1,2,3]. Unfortunately, a number of studies regarding its safe disposal are limited [4,5]. Therefore, there is an urgent need to deal with LZSS in a safe manner. According to the U.S. Environmental Protection Agency (EPA), solidification/stabilization technology has been demonstrated as the best technology for the handling of hazardous and toxic waste.

Basically, harmful/toxic substances are solidified in cementitious materials. In earlier years, Ordinary Portland Cement (OPC) was considered as the best cementitious material which could effectively solidify the heavy metals by precipitation (due to its high pH) and adsorption reactions [6]. Later on, a new type of cementitious material was also named as alkali-activated cementitious material having superior properties such as great resistance to acid attack, low permeability, great resistance to high temperature and long-term durability, which were introduced for solidification/stabilization purposes [6,7,8,9,10]. Additionally, these cementitious materials were more stable to chloride ions exposure as compared to OPC [11]. In general, alkali-activated cementitious materials could be divided into geopolymer and alkali activated slag (AAS) [12]. Geopolymer are raw materials having massive Si and Al (such as fly ash and coal gangue) and AAS refers to BFS (rich Si and Ca), activated by an alkaline solution [12,13]. The BFS is produced in the process of conversion of iron ore to pig iron [14]. The main chemical components of BFS are CaO, SiO_2_, and Al_2_O_3_. The alkaline activation of BFS is carried out in the following manner: (1) Disintegration of the vitreous structure into basic precursors i.e., SiO_4_^4−^, AlO_4_^5−^ and Ca^2+^, (2) recombination: These monomers/basic precursors recombine and form polymer content, (3) condensation: Polycondensation occurs to form a calcium silicate hydrate gel (C-S-H), (4) hardening of gel, which gives rise to a structure with good mechanical properties.

Many studies on the solidification of the hazardous wastes containing heavy metals through AAS are available in the literature. Huang et al. used the BFS-based cementitious material for solidification of chromium slag. They observed that the hexavalent chromium in chromium slag was successfully immobilized without crossing the critical limit, up to 60% addition of chromium slag [15]. In another study, Deja had successfully stabilized the Cr^6+^, Cd^2+^, Zn^2+^ and Pb^2+^ in AAS cementitious material [16]. Similarly, Malolepszy et al. also found that AAS could be used for the solidification of heavy metals [17]. In another study, Gao et al. used the AAS for the preparation of construction material [18]. For this purpose, they used the different ratios of alkali activated cementitious materials and municipal solid waste incineration ash. They found satisfactory leaching results with high compressive strength (20–70 MPa). Taking these studies under consideration, it was concluded that AAS could effectively immobilize the hazardous wastes containing heavy metals.

Bentonite is a common clay mineral which was also used in this experiment as an auxiliary additive for the solidification of LZSS with heavy metals in AAS. As an auxiliary additive, it can fill the pores in solidified bodies and make the structure more compacted with high mechanical strength [19]. In addition to this, bentonite is an economical material which has high water absorption, thickening, high cation exchange capacity and adsorption [20]. Lo cured the contaminated soil in cement matrices and found that the partial replacement of organic bentonite with contaminated soil effectively immobilized the pollutants in solidified bodies, which had high strength and low permeability [21]. It indicated that organic material had a determinantal effect on the hydration process of cement and bentonite reduced the determinantal effect of organic matter. Similar results were also experienced by Conner and Cioffi when they used bentonite as an auxiliary material in cement matrices for the treatment of organic contaminants [22,23]. In another study, Razakamanantsoa et al. also found that the addition of bentonite in soil polymer was beneficial for fixation of pollutants in the soil, which could reduce the permeability coefficient of soil polymer [24]. According to this literature, bentonite was used as an adsorbent in different ways for several kinds of wastes.

According to the best of author knowledge, the bentonite was not used in AAS to enhance the solidification of LZSS. Therefore, the bentonite was utilized in AAS to enhance the compressive strength of solidified products (AASB). In this way, effective solidification and resourceful utilization of LZSS is possible, which was also investigated. 

## 2. Materials and methods

### 2.1. Materials

The BFS (activity coefficient 0.36, quality coefficient 1.63) was brought from a steel plant in Chongqing, China. While, LZSS was collected from Yunnan Chihong Zinc and Germanium Co., Ltd. situated in Yunnan, China. Bentonite used in this experiment was industrial grade. The oxide composition of raw materials is shown in Table 1. The particle size distribution of raw materials is shown in Figure 1. The alkali activators, liquid water glass (silicate modulus 3.3, density 1.46 g cm^−3^) and sodium hydroxide were industrial and analytical grades, respectively.

### 2.2. Methods

#### 2.2.1. The Grinding of Blast Furnace Slag and Lead-Zinc Smelting Slag

The BFS and LZSS were coarser materials. Therefore, both materials were ground separately by a ball mill for 12 h where the speed was set up to 200 rpm to reduce the size and increase the surface area. After grinding, both materials were dried in an oven at 105 °C for 2 h to remove the moisture and then passed through a 200-mesh.

#### 2.2.2. Specimen Preparation

The alkali activator solution was synthesized by mixing water glass (Na_2_O·3.3SiO_2_, industrial grades) and solid sodium hydroxide (NaOH, analytical grades) in deionized water, which was kept at room temperature and used after 18 h. The BFS and alkali activator solutions were mixed well for 5 min and poured in to mold (20 mm × 20 mm × 20 mm). These molds were placed on a shaker for 3 min to remove air bubbles. The specifications of different factors i.e., liquid-to-solid ratio, water glass modulus and temperature (°C) are shown in Figure 2 (single factor experiment, each sample with three replicates (15 × 3 = 45)) and Table 2 (orthogonal experiment, each sample with three replicates (9 × 3 = 27)). The objective of these two experiments was to achieve the highest compressive strength. Compressive strength was evaluated by using similar factors in both experiments, but values of these parameters were re-adjusted around the best values achieved through a single factor experiment and subjected to an orthogonal experiment to achieve more accuracy in the results. The molds were cured at specified temperatures (Figure 2 and Table 2) for 24 h. Afterwards, solidified bodies were unmolded and placed at room temperature for 28 d.

#### 2.2.3. The Synthesis of Alkali Activated Slag Cementitious Material (AAS) with Addition of Bentonite

On the basis of Section 2.2.2, the solidified samples along with bentonite addition (2%, 4%, 6%, 8%, 10%) were synthesized and named as AASB. Furthermore, the compressive strength of AASB with different contents were evaluated.

#### 2.2.4. Solidification of Lead-Zinc Smelting Slag (LZSS)

The different contents (10%, 20%, 30%, 40%, 50%, 60%, 70%) of LZSS were solidified in AAS and AASB by replacing BFS.

#### 2.2.5. Leaching Tests

The leaching toxicity test was conducted using the sulfuric and nitric acid method HJ/T 299-2007, and Toxicity Characteristic Leaching Procedure (TCLP) method. In both methods, solidified bodies were broken down into particle sizes of ≤9.5 mm. In the sulfuric and nitric acid method, the pH of deionized water was adjusted at 3.20 ± 0.05 by mixing sulfuric and nitric acid at a ratio of 2:1. The extractant and solids were mixed in a ratio of 10:1 and shaken for 18 ± 2 h at a rate of 30 ± 2 rpm in a shaker. The TCLP method is also referred to as the U.S. EPA standard and the pH of extractant was adjusted to 2.88 ± 0.05 by adjunction of 5.7 mL of glacial acetic acid. The extractant and solid were mixed in a ratio of 20:1. While, other conditions such as oscillation time and speed were the same as those used in the sulfuric and nitric acid method. The extractants from both methods were pressure filtered and filtrate was used to determine heavy metal concentrations.

#### 2.2.6. Speciation Analysis of Heavy Metal

The full quantitative analysis of heavy metals can characterize the extent of heavy metal impact on the environment, but it cannot truly reflect its potential for ecological hazards. Heavy metals exist in different forms and each form has a varying effect on the ecosystem. Therefore, it is necessary to analyze each fraction of heavy metals after solidification for a better understanding and effectiveness of the solidification process. The universal Tessier progressive extraction method was used to analyze the different fractions of heavy metals i.e., exchangeable, carbonate bounded, iron-manganese oxide bounded, organic and sulfide bounded, and residual [25].

#### 2.2.7. Compressive Strength

The compressive strength of solidified products after 28 d was measured by a universal testing machine (AGN-250, Shimadzu, Japan) according to reference standard GB/T 17671-1999. The value of the compressive strength of the solidified body was averaged from three parallel samples.

#### 2.2.8. The Methods of Analysis

X-ray fluorescence (XRF) was used to analyze the oxide composition of raw materials. X-ray diffraction (XRD) was used to analyze the mineralogical phases of raw materials and solidified products, which was performed with a X’ Pert PRO instrument (PANalytical B.V., Holland). While Fourier Transform Infrared Spectroscopy (FTIR) examination was used to identify the functional groups of raw materials and solidified bodies by using a Nicolet-IS50 (Thermo Fisher Scientific Inc., Waltham, MA, USA) FTIR spectrometer in the range of 4000–400 cm^−1^. The microstructural changes of solidified products were analyzed with a Scanning Electron Microscope (SEM) (Tescan, Mira3 LMH, Czech).

## 3. Results and Discussion

### 3.1. Synthesis of AAS Cementitious Material

#### 3.1.1. Compressive Strength of AAS without Addition of Bentonite

The compressive strength results of single factor and orthogonal experiments are shown in Figure 2 and Table 3. The influence of three factors i.e., water ratios, silica glass modulus and curing temperature on compressive strength were evaluated in both experiments. While an orthogonal experiment based on a single factor experiment was carried out to achieve the accuracy in the final strength. The influence of the aforementioned factors on compressive strength are discussed below.

(a) Single Factor Experiment:

(i) Liquid-to-solid Ratio

Water is a significant parameter influencing the formation of hydration reactant and the structure of the C-S-H gels [26,27]. Hence, different liquid-to-solid ratios were used in the current experiment and is shown in Figure 2a. The compressive strength of AAS increased first and then decreased. While the highest compressive strength (75.83 MPa) occurred at a liquid-to-solid ratio of 0.27. The results indicated that the optimum water contents were necessary for high strength. The lower liquid-to-solid ratio (<0.27) might have resulted in an incomplete reaction process and a higher liquid-to-solid ratio (>0.27) caused the pore formation followed by evaporation with the passage of time. This was also discussed by Yahya [28].

(ii) Water Glass Modulus:

The water glass modulus had a similar trend on compressive strength as observed in the water case. The six different values of water glass modulus were adjusted by the addition of NaOH. It could be observed from Figure 2b that the maximum strength of 75.83 MPa was observed at a water glass modulus of 1.4. The water glass modulus was based on the concentration of silicon which is required in the gelation process. While NaOH is a source of OH^−^ which plays an important role in the disintegration of the BFS structure and ultimately generates the Si, Al and Ca polymers to take part in the reaction process [6,29,30]. The lower water glass modulus (≤1.4) meant higher NaOH contents which resulted in the hinderance of the polycondensation reaction. In contrast, its higher value (>1.4) resulted in a low alkalinity, which hindered the dissolution process and ultimately the incomplete dissolution of BFS had a negative impact on compressive strength. Hence, the appropriate value of the water glass modulus is necessary and was also observed in the current study.

(iii) Curing Temperature:

The curing temperature is also an important parameter regarding solidification of AAS, which directly effects the compressive strength. In the short term, a higher curing temperature might be beneficial for the enhancement of the reaction, but it is not beneficial in the long term because a higher temperature results in the formation of pores due to rapid evaporation of water [31]. Similar results were observed in the current experiment, which showed that compressive strength declined as the curing temperature increased, which can be seen in Figure 2c. The highest compressive strength was obtained at a lower temperature (30 °C). These results were consistent with Muhammad et al. and Kumar et al., who found that the cementitious materials synthesized had the highest compressive strength at 25 °C [26,32].

(b) Orthogonal Experiment

Regarding the orthogonal experiment, the factors with best compressive strength were readjusted e.g., the highest compressive strength, which was 75.83 MPa was obtained at a liquid-to-solid ratio of 0.27 in a single factor experiment, which was then readjusted to 0.26, 0.27, and 0.28 and then finally, compressive strength was determined. In this way, more precise values of all factors could be achieved. The readjusted values based on the single factor experiment are shown in Table 2. In terms of R value (Table 2), the effect of these factors on compressive strength was as follows: Curing temperature (C) > liquid-solid ratio (A) > modulus of water glass (B). According to orthogonal experiment, the highest compressive strength was achieved after curing the cementitious material at 25 °C (C_1_), which was prepared by using a water glass modulus of 1.4 (B_2_) and solid-to-liquid ratio of 0.26 (A_1_). Hence, we could say that the optimal combination of these factors was achieved at A_1_B_2_C_1_.

#### 3.1.2. Synthesis of AAS with Bentonite as an Auxiliary Additive

At a later stage, the different contents of bentonite were added in AAS during its synthesis in accordance with the optimal combination (A_1_B_2_C_1_). The compressive strength of AAS with bentonite is shown in Figure 3 and the samples represent AASB. The slight increase in compressive strength is shown up to a 4% addition of bentonite (on the mass basis of BFS) and further addition of bentonite resulted in a gradual decrease in strength. The increase in strength might have been the result of the bentonite, which served as a filler in the cementitious material due to its small particle size (Figure 1). In contrast to this, bentonite is a highly crystalline material, which is difficult to dissolve in an alkaline solution. Therefore, its high concentration resulted in a decline in strength. According to these results, 4% of bentonite in AAS had a maximum strength which was further utilized for solidification of LZSS (see Section 3.2).

### 3.2. Stabilization/Solidification (S/S) of LZSS

#### 3.2.1. Compressive Strength

The LZSS was solidified in cementitious materials for 28 d, which was optimized through Section 3.1. The amount of BFS was replaced with LZSS up to 70% as shown in Figure 4.

The compressive strength of control samples (solidified bodies without LZSS) was higher as compared to the samples with LZSS which can be seen in Figure 4. Up to 30% addition of LZSS, the compressive strength of solidified bodies having bentonite (AASB) was higher as compared to AAS solidified samples. In general, the compressive strength increased with the addition of LZSS up to a certain amount and then decreased in both kinds of solidified samples i.e., AAS and AASB. The results showed that BFS had a positive interaction with a certain amount of LZSS in both AAS and AASB samples. The further increase in LZSS caused the deterioration of compressive strength due to the increasing content of heavy metals, which had a hindrance effect on the gelation process. This was also described by Palomo et al. [33]. In spite of this, the solidified bodies had a high compressive strength (>60 MPa) even the amount of LZSS was 70%, which showed the potential application of these samples for cleaner production. For example, when utilized on construction (>10 MPa) [34,35]. Alwaeli et al. [4] also experienced the same results where the addition of LZSS up to certain level in a cement mixture had a positive effect on compressive strength when it was used for construction purposes.

#### 3.2.2. Leaching Experiment

The results of heavy metal leaching behavior in the original LZSS samples are shown in Table 3.

According to HJ/T 299-2007 (sulfuric and nitric acid method), the heavy metals had not exceeded their toxicity limits. While Pb exceeded the toxicity limits as specified by EPA standards. Additionally, the leaching concentration of Zn (368.60 mg/L) was too high in the TCLP method. The leaching concentrations of Zn and Pb formed solidified bodies containing LZSS and are shown in Figure 5. Generally, higher leaching concentrations were observed with the TCLP method as compared to HJ/T 299-2007 (sulfuric and nitric acid method). Additionally, as can be seen in Figure 5, leaching concentrations of Pb and Zn were almost consistent with compressive strength such that the heavy metals leaching concentration increased as the compressive strength decreased (Figure 4) after 30% and 50% addition of LZSS in AASB and AAS samples, respectively. Additionally, the leaching concentration of Pb and Zn from solidified bodies was much lower than raw LZSS and had not exceeded the critical limits defined by EPA and GB5085.3-2007 standards. Overall, AASB had less leaching or a higher solidification effect of Pb and Zn as compared to AAS due to the presence of bentonite which has high adsorption capacity for heavy metals. On the other hand, lower leaching of Pb as compared to Zn was due to its smaller hydrated ion and higher affinity towards adsorption, substitution and precipitation. It was also explained by Cheng et al. [36]. Huang et al. and Li et al. explained that these metal cations were solidified through physical adsorption, ion exchange, surface complexation and precipitation [34,37]. Regarding Pb solidification, these mechanisms were explained by Thevenin and Pera, which are presented in Equations (1) to (3) [38].
Adsorption: C-S-H + Pb → Pb-C-S-H(1)
Substitution: C-S-H + Pb^2+^ → Pb-S-H + Ca^2+^(2)
Precipitation: Pb^2+^ + OH^−^ + Ca^2+^ + SO^4−^ → mixed salts(3)

Conclusively, it was found that the solidification effect depended upon the composition of the binder material, type of leaching extractant, nature and concentration of waste (especially in the context of heavy metals).

### 3.3. Heavy Metals Speciation Analysis

Heavy metal speciation analysis was performed on AAS and AASB with 70% LZSS, and the results are shown in Figure 6.

It can be seen from the Figure 6 that exchangeable, carbonate and Fe-Mn oxide bounded fractions of Zn and Pb were lower in AASB as compared to AAS. The significant reduction of exchangeable and carbonate bounded Zn (AAS = 43.35% and AASB = 47.37%) and Pb (AAS = 47.03% and AASB = 51.78%) was observed after solidification. As compared to AAS, the lower proportion of highly mobile fractions (exchangeable and carbonate bounded) in AASB samples indicated that bentonite had a stronger effect on the stabilization of Zn and Pb. Moreover, Fe-Mn oxide bounded fraction also reduced in both samples (AAS and AASB) after solidification as compared to the raw material (LZSS). In contrast, the proportion of the organic and sulfide bounded forms had not varied, and the residual fraction greatly increased. It was concluded that heavy metals exist in a more stable form due to chemical fixation. Obviously, bentonite improved the solidification effect of heavy metals (Pb and Zn) in LZSS by chemical reaction.

### 3.4. Material Characterization

#### 3.4.1. The X-ray Diffraction (XRD) Analysis

The XRD patterns of raw materials are shown in Figure 7. The BFS is an amorphous material which could be observed by the presence of an amorphous hump in the range of 20° and 40° at 2θ. In addition to this, a small amount of gehlenite (PDF#35-0755) was also observed. The bentonite was highly crystalline, which was mainly composed of montmorillonite (PDF#29-1498) and quartz (PDF#46-1045). The absence of crystalline structures in the XRD pattern of LZSS showed that it was amorphous in nature and two humps were observed in the range of 10°–20° and 25°–40° at 2θ. While, the XRD pattern of AAS50% and AASB50% (Figure 8) indicated that they were also amorphous materials instead of crystalline materials. Some studies have proven that C-S-H gel is the major hydration product of AAS when the SiO_2_/Al_2_O_3_ is below 4.5 [39] According to these results, no crystalline phases were observed in solidified bodies because BFS phases were dissolved after alkaline activation and a small amount of bentonite was produced. It also showed that bentonite had not affected the structure but acted as an adsorbent for heavy metal ions and also as a filler to improve the strength.

#### 3.4.2. The Fourier Transform Infrared Spectroscopy (FTIR) Analysis

The FTIR analysis of samples are shown in Figure 9. In all samples, the stretching vibration of O-H at 1640 cm^−1^ to 1650 cm^−1^ and the bending vibration of H-O-H at 3300 cm^−1^ to 3500 cm^−1^ were correspondent to water molecules, which was also confirmed by Bernal et al. and Lee et al. [8,40,41]. While the weak peaks around 3500 cm^−1^ to 4000 cm^−1^ were characterized as hydrated aluminum compounds [42]. The CO_2_ from air reacted with alkali metal oxide, which was pointed out by the bending vibration of O-C-O at 1450 cm^−1^ [43,44,45]. The peaks at 650 cm^−1^ to 660 cm^−1^ were a symmetric vibration of Al-O-Si and Si-O-Si bonds [40]. The weak peaks between 400 cm^−1^ to 500 cm^−1^ resulted from the bending vibration of Si-O-Si and O-Si-O bonds [42]. The wavenumber of 940 cm^−1^ was correspondent to a stretching vibration of Si-O-Si or Si-O-Al, which resulted from the hydration reaction indicating that major hydration reaction products were C-S-H phases [46]. The wavenumber of Si-O-Si (Al) moved towards a lower value after the addition of LZSS had indicated that heavy metals in LZSS were attached to the C-S-H gel and ultimately influenced the bonds in the gel structure. Generally, Al^3+^ is surrounded by four oxygen atoms which gives rise to a net negative charge by forming [AlO_4_]. Moreover, Ca^2+^ and Na^+^ in structure are used as charge balancing ions. These charge balancing ions might be replaced by heavy metal cations and encapsulated/solidified in the cementitious structure. A similar mechanism was also explained by Muhammad F et al. and Wang et al. [26,47]. In addition, the increase in peak intensity at approximately 940 cm^−1^ (in the case of AASB 50% as compared to AAS 50%) was observed, indicating that bentonite might promote the information of C-S-H.

#### 3.4.3. The Scanning Electron Microscope Analysis

In order to further explore the solidification mechanism, the SEM of raw materials (BFS and LZSS) and solidified bodies (AAS, AAS 50% and AASB 50%) was carried out. The SEM images of BFS and LZSS verified that these materials were amorphous in nature because they had irregular shapes with sharp edges and corners which can be seen in Figure 10a,b. While AAS (Figure 10c,d) had no sharp edges and corners which showed that BFS was dissolved, polymerized and formed hardened gel after alkaline activation. It could also be seen that AAS had a dense structure, which was consistent with compressive strength. The SEM images of AAS 50% and AASB 50% are shown in Figure 10e,f respectively. This indicated that the addition of LZSS resulted in the formation of a few holes and cracks, which was the main reason for a decreased compressive strength compared with the control solidified body (without LZSS). In spite of this, the structure of solidified bodies was dense enough, which hindered the leaching of heavy metals.

## 4. Conclusions

In this paper, a single factor and orthogonal experiment on three factors (water glass modulus, liquid-solid ratio and curing temperature) was designed to evaluate the compressive strength of AAS. The highest compressive strength was achieved by using a water glass modulus of 1.4, liquid-to-solid ratio of 0.26 and curing temperature of 25 °C. Moreover, the samples with the highest compressive strength were also prepared by substituting different proportions of bentonite and a higher compressive strength was observed with the 4% addition of bentonite. Afterwards, the samples (AAS and AASB) with the highest compressive strength were used to solidify LZSS which had hazardous heavy metals i.e., Pb and Zn. Overall, the alkali-activated solidified bodies had satisfactory compressive strength (>60 MPa, up to 70% addition of LZSS). At a lower level of LZSS (≤30%), the positive interaction of BFS and LZSS resulted in an increase in the compressive strength of solidified bodies. However, the decrease in strength with the addition of LZSS (>40%) was due to less activity substance in cementitious material and a higher concentration of heavy metals, which hindered the hydration reaction. The speciation analysis of heavy metals showed that easily releasable fractions (exchangeable and carbonate bounded) of heavy metals reduced after solidification. Moreover, the heavy metals during leaching experiments did not exceed the critical limits defined by EPA and GB5085.3-2007 standards. As expected, a better solidification effect was observed in AASB due to the presence of bentonite, which had a higher adsorption capacity. These solidified bodies with higher compressive strength and lower leaching concentration could be used for both landfill and construction purposes. Regarding the solidification mechanism, the XRD results indicated that crystalline structures were dissolved and no crystalline materials with heavy metals were generated which had proven that heavy metals in LZSS were adhered to the surface of C-S-H due to adsorption and complexation as well as encapsulated by a network structure of cementitious material. The FTIR results indicated that the heavy metals in LZSS influenced the hydration reaction in solidified bodies which resulted in the shift in Si-O-Si (Al) towards a lower wavenumber and the degree of hydration reaction could be improved in solidified bodies with bentonite. The SEM results showed that the internal structure of AAS and solidified bodies were dense, which was beneficial for the solidification of LZSS. On the basis of the analysis of XRD, FTIR and SEM, the solidification of heavy metals in LZSS was carried out by both physical encapsulation and chemical fixation. Additionally, bentonite as an auxiliary additive in AAS obviously improved the solidification effect of LZSS by increasing the degree of the hydration reaction of AAS and enhanced the chemical adsorption capacity of heavy metals in LZSS.

## Figures and Tables

**Figure 1 ijerph-16-01121-f001:**
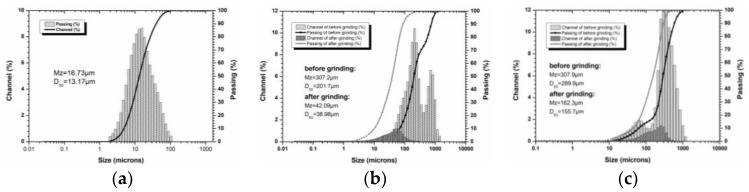
The particle size distribution of raw materials (**a**) bentonite; (**b**) Blast Furnace Slag; (**c**) Lead-Zinc Smelting Slag.

**Figure 2 ijerph-16-01121-f002:**
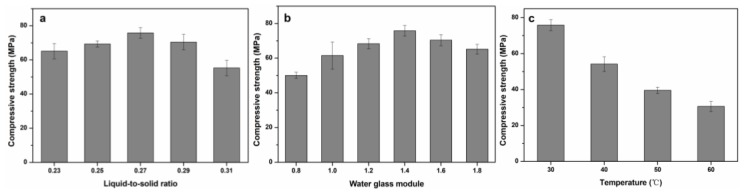
Effect of different factors on compressive strength in single factor experiments (**a**) Liquid-to-Solid Ratio; (**b**) Water Glass Modulus; (**c**) Temperature.

**Figure 3 ijerph-16-01121-f003:**
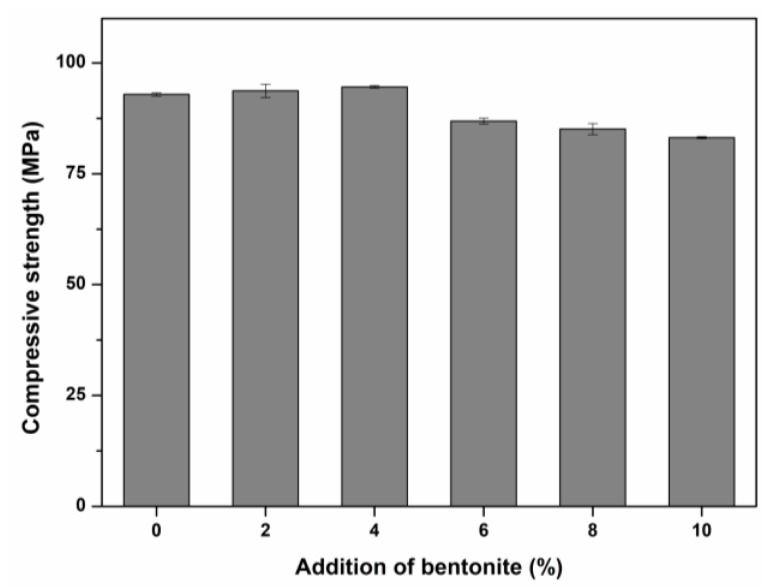
The effect of bentonite on compressive strength.

**Figure 4 ijerph-16-01121-f004:**
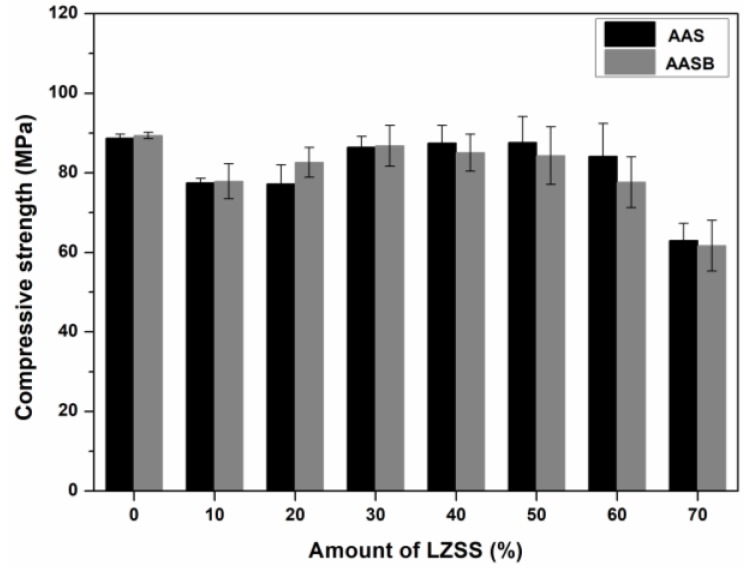
Compressive strength of solidified bodies with LZSS.

**Figure 5 ijerph-16-01121-f005:**
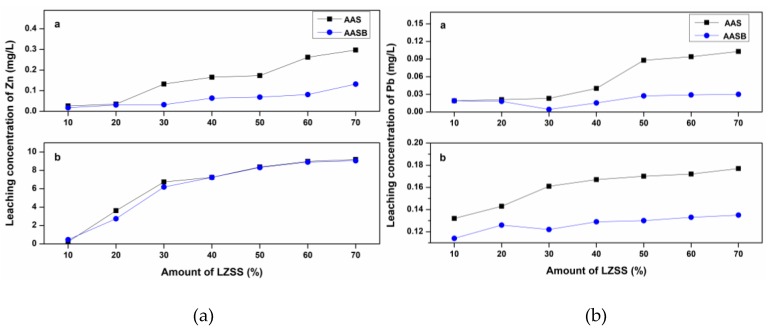
The leaching concentration of (**a**) Zn and (**b**) Pb (**a**- sulfuric and nitric acid method; **b**-toxicity characteristic leaching procedure (TC LP) method).

**Figure 6 ijerph-16-01121-f006:**
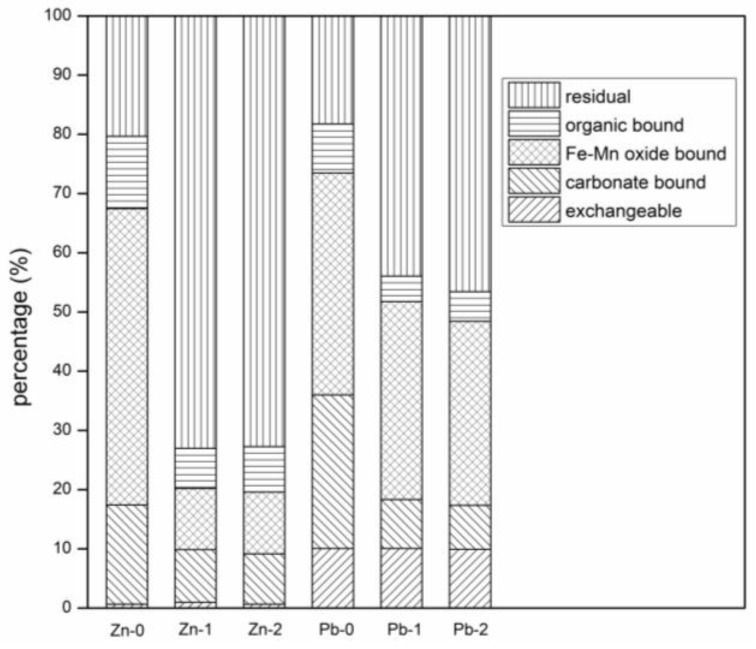
Results of morphological analysis (0-LZSS; 1-AAS; 2-AASB).

**Figure 7 ijerph-16-01121-f007:**
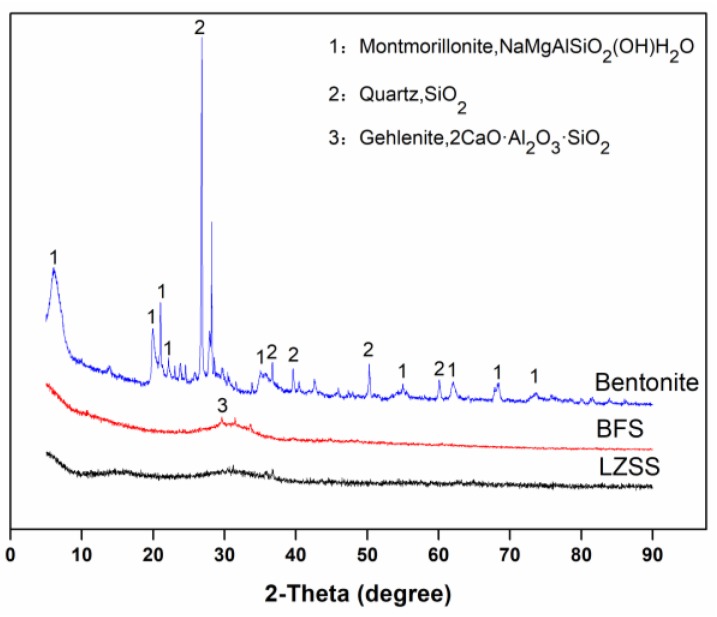
The XRD patterns of raw materials.

**Figure 8 ijerph-16-01121-f008:**
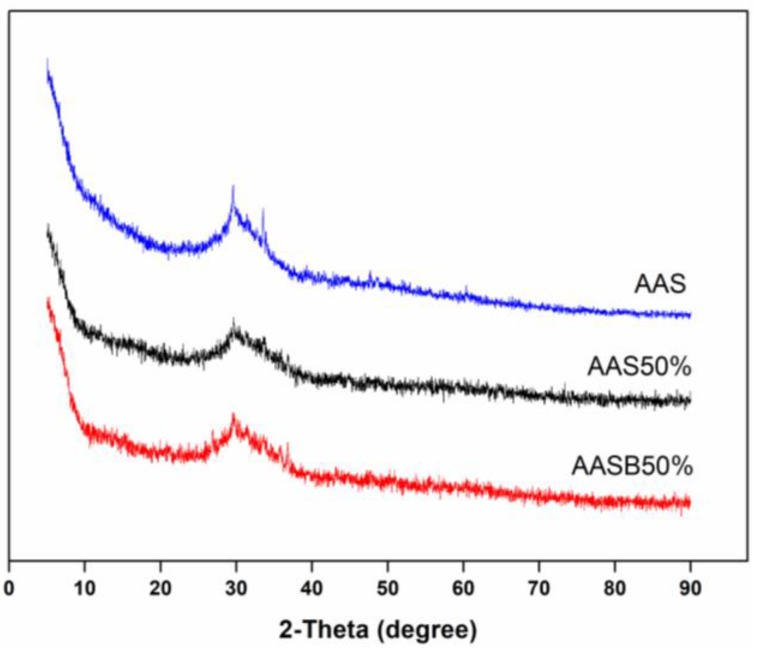
The XRD patterns of AAS cementitious material (containing LZSS).

**Figure 9 ijerph-16-01121-f009:**
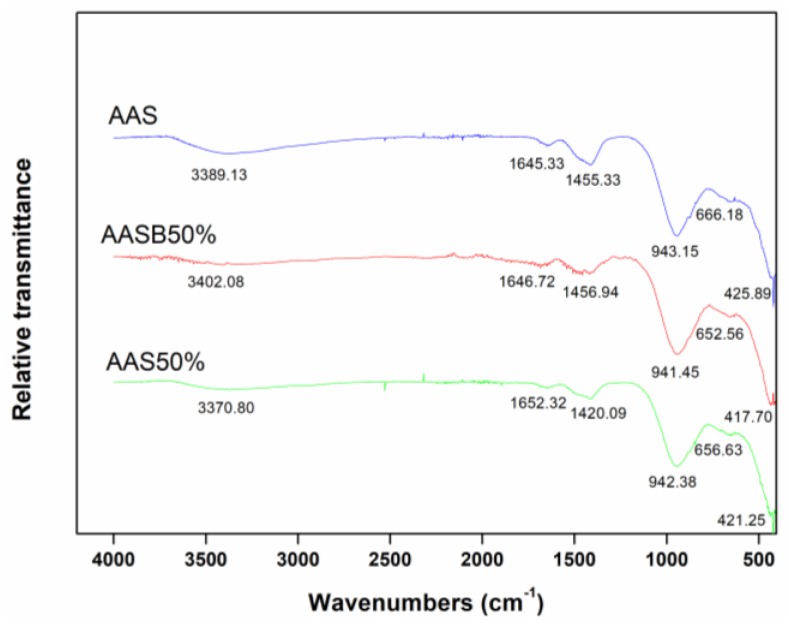
The FTIR spectra of LZSS and the solidified body.

**Figure 10 ijerph-16-01121-f010:**
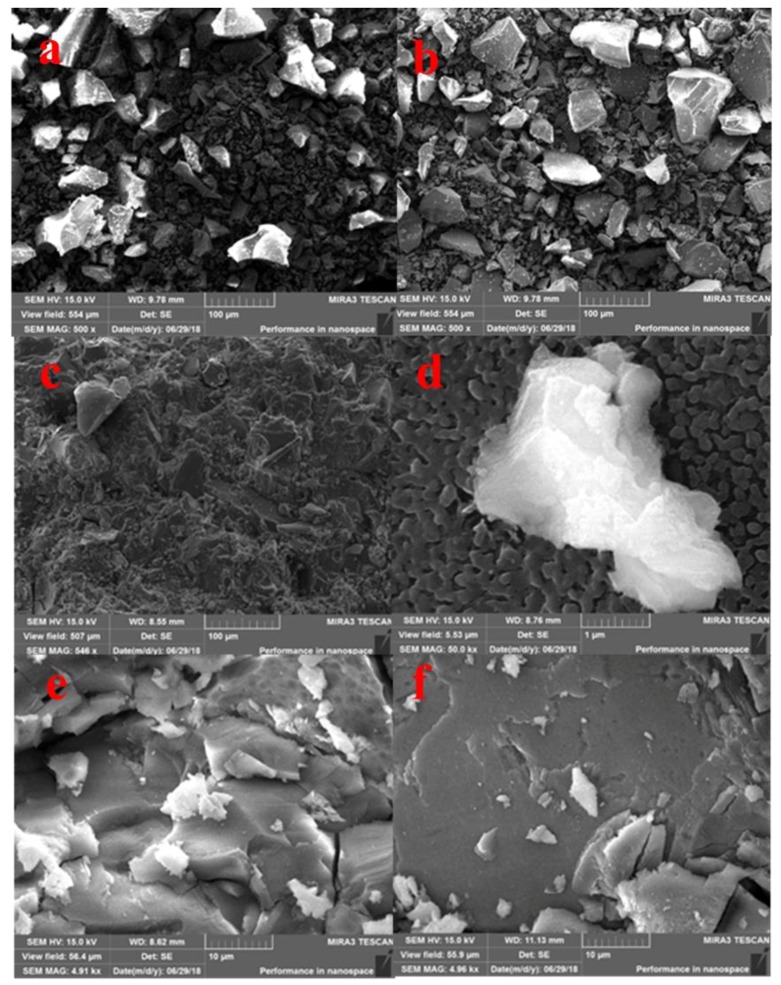
The SEM images of (**a**) BFS (**b**) LZSS (**c**)AAS (**d**) AAS (**e**) AAS50% (**f**) AASB50%.

**Table 1 ijerph-16-01121-t001:** The oxide composition of raw materials (Weight (%)).

Material	CaO	SiO_2_	Al_2_O_3_	MgO	TiO_2_	SO_3_	Fe_2_O_3_	K_2_O	Na_2_O	P_2_O_5_
BFS	40.43	30.18	10.77	7.91	6.01	3.21	0.64	0.56	0.28	-
LZSS	12.48	30.67	7.27	3.27	0.90	3.06	29.73	0.96	0.72	0.15
Bentonite	2.59	70.52	15.85	2.09	0.51	0.04	3.34	1.07	3.61	0.08
Material	ZnO	MnO	CuO	Cr_2_O_3_	PbO	NiO	BaO	SrO	ZrO_2_	-
BFS	-	-	-	-	-	-	-	-	-	-
LZSS	6.39	2.97	0.55	0.12	0.03	0.02	0.14	0.41	0.04	-
Bentonite	0.01	0.11	-	-	-	-	0.08	0.05	0.04	-

**Table 2 ijerph-16-01121-t002:** The layout and statistical analysis of the orthogonal experiment.

No.	A (Liquid-to-Solid Ratio)	B (Water Glass Modulus)	C (Curing Temperature)	Compressive Strength/(MPa)
1	1(0.26)	1(1.3)	1(25 ℃)	99.39
2	1	2(1.4)	2(30 ℃)	84.99
3	1	3(1.5)	3(35 ℃)	68.16
4	2(0.27)	1	2	80.35
5	2	2	3	73.15
6	2	3	1	95.76
7	3(0.28)	1	3	60.09
8	3	2	1	87.46
9	3	3	2	79.28
K_j1_	252.56	239.83	282.62	
K_j2_	249.26	245.60	244.62	
K_j3_	226.82	243.20	201.40	
k_j1_	84.19	79.94	94.21	
k_j2_	83.09	81.87	81.54	
k_j3_	75.61	81.07	67.13	
Optimal level	A_1_	B_2_	C_1_	
R	8.58	1.93	27.07	
Order	C > A > B			
Optimal combination	A_1_B_2_C_1_			

**Table 3 ijerph-16-01121-t003:** The leaching concentrations of raw LZSS and their critical limits.

Heavy Metal	Cr (mg/L)	Zn (mg/L)	Pb (mg/L)	Cu (mg/L)	Ni (mg/L)	Mn (mg/L)
TCLP method	0.59	368.60	7.50	1.48	0.56	71.83
sulfuric and nitric acid method	0.07	5.23	0.15	0.13	0.27	3.76
Toxicity limitation (US EPA) standard	5	-	5	15	-	-
Toxicity limitation (GB5085.3-2007).	15	100	5	100	5	-

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
