# Peer review of "The Solidification of Lead-Zinc Smelting Slag through Bentonite Supported Alkali-Activated Slag Cementitious Material"

_ijerph, 2019, doi:10.3390/ijerph16071121_

Round 1

Reviewer 1 Report

This article is persuasive and well-written. To improve the quality of the artcle, it is necessary to make some corrections.

 In subchapter 2.1, add physical data of each materials to enhance the understandability please. 

In 2.2.2 Specimen preparation, number of sepcimens is to be added.

In tables 2 and 3, it is necessary to provide the figure, instead of table.

Author Response

Dear reviewers,

We would like to express our sincere appreciation for your careful reading and helpful comments.

Those comments are all valuable and very helpful for revising and improving our paper, as well as

the important guiding significance to our researches. We have addressed the points.

Responses to Reviewer #1:

Comment: This article is persuasive and well-written. To improve the quality of the article, it is necessary to make some corrections.

In subchapter 2.1, add physical data of each materials to enhance the understandability please.

In 2.2.2 Specimen preparation, number of specimens is to be added.

In tables 2 and 3, it is necessary to provide the figure, instead of table.

Response: Firstly, thank you very much for your kind appreciation. Your comments are really valuable. Now, we have modified it according to your guidelines and we hope these will be satisfactory.

Point 1: In subchapter 2.1, add physical data of each materials to enhance the understandability please.

Response 1: Thank you for your valuable comments. Now we have added the particle size distribution of the raw materials (page: 3, line: 40-41; page: 4, line: 1-3) and coefficients of activity and quality of BFS (page: 3, line:37).

Point 2: In 2.2.2 Specimen preparation, number of specimens is to be added.

Response 2: Thank you for your advice. The number of specimens have been added in section 2.2.2 (page: 4, line: 19-20).

Point 3: In tables 2 and 3, it is necessary to provide the figure, instead of table.

Response 3: Thank you for your valuable comments. According to your comments, we have added the figure (named Fig.2) about single factor experiments instead of table 2 (page: 6, line: 11-12). Regarding table 3, it shows the layout and statistical analysis of orthogonal experiment. The statistical analysis is based on the data of orthogonal experiment which is shown in table 3. However, the figure can only describe the result of compressive strength. So, we keep table 3 (named table 2 in the revised manuscript).

At last, thank you for your valuable advice.

Reviewer 2 Report

The paper discusses interesting topic of stabilization of lead-zinc smelting slag. The paper covers a lot of experimental results. However, the overall content of the paper is confusing. The main problem is the poor language and style. The text is not concise, the order of the paragraphs is confusing and unrelevant things are mentioned, particularly in the Introduction.

Examples: “Numerous waste materials having aluminosilicate precursors such as Blast Furnace Slag (BFS)”

“In current experiment, LZSS was solidified through BFS due to its low porosity which resulted in formation of denser microstructures and ultimately have better durability characteristics [14].”

Very unclear part on page 5, lines 12-17.  

Also the conclusions are a bit too far gone, for example: "Furthermore, no new crystalline structures associated with Zn or Pb had formed which meant that these metals had become the part of C-S-H gels." 

The stabilization and compressive strength results itself are good, but do not provide much new information about the stabilization or reaction mechanisms. 

Author Response

Dear reviewers,

We would like to express our sincere appreciation for your careful reading and helpful comments.

Those comments are all valuable and very helpful for revising and improving our paper, as well as

the important guiding significance to our researches. We have addressed the points.

Responses to Reviewer #2:

Comment: The paper discusses interesting topic of stabilization of lead-zinc smelting slag. The paper covers a lot of experimental results. However, the overall content of the paper is confusing. The main problem is the poor language and style. The text is not concise, the order of the paragraphs is confusing and irrelevant things are mentioned, particularly in the Introduction.

Examples: “Numerous waste materials having aluminosilicate precursors such as Blast Furnace Slag (BFS)”

In current experiment, LZSS was solidified through BFS due to its low porosity which resulted in formation of denser microstructures and ultimately have better durability characteristics [14].”

Very unclear part on page 5, lines 12-17.

Also the conclusions are a bit too far gone, for example: "Furthermore, no new crystalline structures associated with Zn or Pb had formed which meant that these metals had become the part of C-S-H gels."

The stabilization and compressive strength results itself are good, but do not provide much new information about the stabilization or reaction mechanisms.

Response: Firstly, thank you very much for your kind appreciation. Your comments are really valuable. Now, we have modified it according to your guidelines and we hope these will be satisfactory.

Point 1: The text is not concise, the order of the paragraphs is confusing and irrelevant things are mentioned, particularly in the Introduction.

Examples: “Numerous waste materials having aluminosilicate precursors such as Blast Furnace Slag (BFS)”

In current experiment, LZSS was solidified through BFS due to its low porosity which resulted in formation of denser microstructures and ultimately have better durability characteristics [14].”

Response 1: Thank you for your valuable comments. The modifications have been done according to your valuable comments. We have revised the paper to be more concise and adjusted some content (page: 2, line: 14-51; page: 3, line: 20-34; page: 4, line: 27-30; page: 5, line: 1-3; page: 7, line: 15; page: 8, line: 32-38; page: 9, line: 13-21; page: 10, line: 1-8; page: 10, line: 22-24; page: 12, line: 4-6; page: 12, line: 24-26; page: 13, line: 11-15; page: 13, line: 21-22; page: 13, line: 26-28).

Point 2: Very unclear part on page 5, lines 12-17.

Response 2: We are sorry for that the sentence is presented unclear. Now we have modified it and make it clear (page: 6, line: 14-18).

Point 3: Also the conclusions are a bit too far gone, for example: "Furthermore, no new crystalline structures associated with Zn or Pb had formed which meant that these metals had become the part of C-S-H gels."

Response 3: Thank you for your valuable comments. We have deleted something inappropriate and revised the conclusion section seriously (page: 16, line: 4-40).

Point 4: The stabilization and compressive strength results itself are good, but do not provide much new information about the stabilization or reaction mechanisms.

Response 4: Thank you for your valuable comments. Now we have cited relevant literature and combined the analysis of XRD, FTIR and SEM to add some stabilization mechanisms in the section 3.4.1, 3.4.2 and 3.4.3 (page: 12, line:20-24; page: 13, line:17-28; page: 14, line: 17-18). In addition, some corresponding supplements have been made in conclusion (page: 16, line: 20-37).

At last, thank you for your suggestion.

Round 2

Reviewer 2 Report

Thank you for providing detailed list of revised sections. 

The main concern is still the XRD section and the conclusions drawn from it. It is well reported that in AAS the main binder phase is C(N)ASH with often Mg-Al LDHs as secondary phases [Rilem State-of-the-Art report, Alkali-activated materials, Provis, van Deventer] with broad peak around 30 degree 2 theta which is visible in results presented. Maybe some CaCO3 also there, but hard to say. 

The reference in XRD section for CSH identification ([39]: A review on alkali-aggregate reactions in alkali-activated mortars/concretes made with alkali-reactive aggregates]) is an odd choice as the reference is not really related to XRD identifications.